# Physicochemical Properties of Mesoporous Organo-Silica Xerogels Fabricated through Organo Catalyst

**DOI:** 10.3390/membranes11080607

**Published:** 2021-08-10

**Authors:** Muthia Elma, Anna Sumardi, Adhe Paramita, Aulia Rahma, Aptar Eka Lestari, Dede Heri Yuli Yanto, Sutarto Hadi, Zaini Lambri Assyaifi, Yanuardi Raharjo

**Affiliations:** 1Department of Chemical Engineering, Faculty of Engineering, Lambung Mangkurat University, Jl. A. Yani KM 36, Banjarbaru 70714, Indonesia; annasumardi75@gmail.com (A.S.); adheprm23@gmail.com (A.P.); aptarlestari97@gmail.com (A.E.L.); zaini.lambri.assyaifi@gmail.com (Z.L.A.); 2Materials and Membranes Research Group (M2ReG), Lambung Mangkurat University, Jl. A. Yani KM 36, Banjarbaru 70714, Indonesia; arahma@mhs.ulm.ac.id; 3Research Center for Biomaterials, Indonesian Institute of Sciences, Jl. Raya Bogor Km.46, Cibinong Science Center, Cibinong, Bogor 16911, Indonesia; dede@biomaterial.lipi.go.id; 4Mathematics Department, Faculty of Teacher Training and Education, Lambung Mangkurat University, Banjarmasin 70123, Indonesia; sutarto.hadi@ulm.ac.id; 5Department of Chemistry, Faculty of Mathematic and Natural Sciences, Lambung Mangkurat University, Jl. A. Yani KM 36, Banjarbaru 70714, Indonesia; sunardi@ulm.ac.id; 6Membrane Science and Technology Research Group, Chemistry Department, Faculty of Science and Technology, Universitas Airlangga, Surabaya 60115, Indonesia; yanuardiraharjo@fst.unair.ac.id

**Keywords:** organo-silica xerogel, mesoporous material, one-step catalyst, two-step catalyst

## Abstract

The physicochemical properties of organo-silica xerogels derived from organo catalyst were pervasively investigated, including the effect of one-step catalyst (citric acid) and two-step catalyst (acid-base), and also to observe the effect of sol pH of organo-silica xerogel toward the structure and deconvolution characteristic. The organo-silica xerogels were characterized by FTIR, TGA and nitrogen sorption to obtain the physicochemical properties. The silica sol–gel method was applied to processed materials by employing TEOS (tetraethyl orthosilicate) as the main precursor. The final molar ratio of organo-silica was 1:38:x:y:5 (TEOS:ethanol: citric acid: NH_3_:H_2_O) where x is citric acid concentration (0.1–10 × 10^−2^ M) and y is ammonia concentration (0 to 3 × 10^−3^ M). FTIR spectra shows that the one-step catalyst xerogel using citric acid was handing over the higher Si-O-Si concentration as well as Si-C bonding than the dual catalyst xerogels with the presence of a base catalyst. The results exhibited that the highest relative area ratio of silanol/siloxane were 0.2972 and 0.1262 for organo catalyst loading at pH 6 and 6.5 of organo-silica sols, respectively. On the other hand, the organo-silica matrices in this work showed high surface area 546 m^2^ g^−1^ pH 6.5 (0.07 × 10^−2^ N citric acid) with pore size ~2.9 nm. It is concluded that the xerogels have mesoporous structures, which are effective for further application to separate NaCl in water desalination.

## 1. Introduction

Materials of mesoporous structure are synthesized by sol–gel process, which is a versatile approach to form functional materials for membranes, sensors, catalytic and optical applications. In the past few years, fabrication and application of thin film as a membrane for separation have become a concern to development, especially for desalination. There are two types of materials commonly used such as organic and inorganic. Organic-based materials such as polymers are widely utilized for water purification, and inorganic for gas separation. However, inorganic-based materials are offering more advantages, i.e., robustness, high molecular sieving, resistance to high temperature and long lifespan. Silica is one inorganic based material that has good chemical stability and is affordable to be employed for preparation xerogel. This is due to their corresponding porosity, surface area [1,2,3,4] and morphological control [5,6,7].

Various formulation of sol–gel processes has been utilized by researcher to fabricate a high-quality membrane. Raman et al. [8] demonstrated the purpose of an organic templated rapprochement for membrane synthesis with good pore design ability. The study reported the use of TEOS and MTES as silane precursors with HCl as a catalyst in the sol–gel preparation for intermediate layers for gas permeation.

Pure silica membrane-derived TEOS synthesis by two-step (nitric acid-ammonia) catalyst has been studied by Elma et al. [9]. Although this material produces a mesoporous structure that is good for desalination, the membrane performance still decreases. The decrease of salt rejection for the pure silica membranes and changes in the flux regime strongly suggests that pure silica films are not hydro-stable. These membranes have a large concentration of silanol (Si-OH) groups, which are hydrophilic in behavior. As water exposes with silanols, it causes the silica to become mobile.

To address the hydro-stability of porous silica matrices, several functional groups have embedded structural stabilizing unities into the silica matrix. These have included carbonization of cationic surfactants, hybrid with carbon/surfactant [10,11,12], covalently bonded templates, and doping of metal oxides based on cobalt [13,14,15].

Wijaya et al. [16] reported the manufacture of carbonized template silica (CTS) membrane by two-step acid catalyst using nitric acid, TEOS and surfactants C16 (hexadecyltrimethylammonium bromide). These works obtain high surface BET of 793 m^2^ g^−1^ and pore volume 0.37 cm^3^ g^−1^. The CTS membranes are hydro-stable, though flux tends to vary as a function of time. However, the surfactant of C16 is very costly and is not a renewable material. It is also similar to another study published by Yang et al. [17] employing P123.

There are several studies to overcome the costly and environmentally friendly issues as an option material for manufacturing carbon-silica-based membrane. Pectin is one of carbon source that can be used for fabrication of carbon-silica membrane [18]. Preparation of silica-pectin membrane was performed by two-step (nitric acid-ammonia) catalyst and a template of pectin from apple peel. The performance of membrane shows excellent high water flux over 7 kg m^−2^ h^−1^ for brackish water desalination. However, template strategies meant that the membrane fabrication spent more time above 6 h. Due to that, it may be necessary to apply hybrid strategies to shorten the production membrane time under 6 h.

Chua et al. [19] reported that hybrid membrane was prepared under a two-step catalyst using HCl as an acid catalyst, BTESE as a precursor and carbon as a surfactant. Its membrane has a high surface area BET, pore volume and pore size of 310 m^2^ g^−1^, 0.18 cm^3^ g^−1^ and 2 nm, respectively. However, this membrane still has limitations due to salt deposits on the permeate side of membrane surface at high saline feed water, which has a higher risk of pore wetting. Therefore, the membrane properties and surface chemistry need to be well designed and balanced to ensure a high permeation flux, perfect salt rejection and no pore wetting within the system.

The study published by Sumardi et al. [20] investigated a mesoporous hybrid organo-silica thin film from organic catalyst. Instead of using TEOS as a silane precursor, this research combined dual precursor with TEVS to enrich the carbon into silica matrices. Moreover, the utilization of organo catalyst (e.g., citric acid) also has a role as a carbon source and to control the pore size structure, whereas the use of single precursor by TEOS and organo catalyst for the fabrication of organo-silica membrane is not the focus of the investigation.

The above review shows that conventionally high-quality silica-based membranes have been synthesised by sol–gel processes mainly including a single-step catalysed hydrolysis of TEOS using HNO_3_ as a catalyst, or a templated single-step, or templated two-step using HCl followed by pore tuning processes. The employment of a two-step catalysed hydrolysis process has also been reported to prepare high-quality silica-based membranes. The two-step hydrolysis condensation of silicon polymers results in the formation of weakly branched systems [21]. For weakly branched systems, there is a higher tendency for structures to interpenetrate, forming large structures of micropore size, resulting in densification and apertures of molecular sieve dimensions.

The purpose of this study was to employ citric acid to produce a silica thin film using one- and two-step organic acid-base catalysts for the sol–gel process, which is very popular in the membrane technology field. Other than that, tailoring of pore size of thin film depended on silica sol pH values. The decrease of the pH determines the shrinking of the pore size in the silica matrices to micropores [7]. Therefore, it requires to ensure the sol pH is varied by the addition of different citric acid concentrations.

This work will show that one of important features of the one- and two-step sol–gel processed organo-silica membranes is greater pore size tailor ability allowing superior desalination performance over conventional one- and two-step with template or hybrid dual precursor processed membranes. Hence, the fundamental measures will be used to test the novel features of this study. First, we will investigate the structural characteristics of the organo-silica xerogels based on FTIR, N_2_ physisorption correlated to surface structure and pore size, and TG analysis, which can be correlated to the surface-functionalization mechanism in organo-silica xerogels.

## 2. Materials and Methods

### 2.1. Chemical and Materials

Several materials and chemicals have been employed for this work, i.e., tetraethyl orthosilicate (TEOS, 99.0%, (GC) Sigma-Aldrich, St. Louis, MO, USA) as silica precursor, ammonia solution (NH_3_, 25%, Merck, Darmstadt, Germany), diluted citric acid (0.001 M C_6_H_8_O_7_), ethanol (EtOH, 99%) as a solvent and demineralized water. The organo-silica sol–gel set-up was described in Figure 1.

### 2.2. Synthesis of Sol Gel Process

The silica sol was synthesized by a simple sol–gel technique, of which the following detail procedure refers to our previous work [22,23]. Firstly, silica sol via two-step catalyst was prepared by mixing TEOS and ethanol for 5 min in an ice bath at 0 °C with 250 rpm; subsequently, the diluted citric acid with demineralized water was dropwise, and the reagent bottle was moved from an ice bath into a water bath and subsequently reflux for an hour at 50 °C with mixing speed 2500 rpm. Afterward, the diluted ammonia with ethanol was added dropwise within 20 min into the solution for 2 h by mixing it in similar conditions. The final pH of organo-silica sol formed was left cold and measured using a pH meter. Meanwhile, the one-step catalyst was prepared in a similar way, but without ammonia, the reflux time became 3 h.

The reagent bottle was prepared and submerged into a bowl as shown in Figure 1a in an ice bath at 0 °C; the second condition is under heating at 50 °C using a water bath, as shown in Figure 1b. The various organo-silica sol pH measurements were prepared similar to the two-step catalyst procedure with various citric acid concentrations. The final molar ratio of multiple organo-silica xerogels was listed in Table 1.

### 2.3. Preparation and Characterization of Organo-Silica Xeorgel

The obtained organo-silica sol was following placed in the Petri dish and dried in oven at 60 °C for 24 h. Hereinafter, the dried sol was named organo-silica xerogel. The organo-silica xerogel was grounded to a powder and calcined at 200 °C using a furnace under air condition by the RTP (rapid thermal processing) technique for 1 h without applying ramping/cooling rates.

FTIR (Fourier transform infra-fed) is used to investigate the chemical properties of silica-carbon xerogels. FTIR spectra data were collected from FTIR type Bruker Alpha. Instrument type: alpha sample compartment RT-DLaTGS accessory: ATR platinum Diamond 1 Relf. The spectra were collected from a total of 30 scans ranging between wavelengths of 600–4000 cm^−1^. Peak deconvolution of the absorption bands over the region 1300–700 cm^−1^ was performed with Fityk software using Gaussian line shapes with a least square fit routine [24] and peak areas were measured for the normalized spectra using a local baseline. Nitrogen physisorption analysis at 77 K and 1 bar were conducted using a Micromeritics TriStar 3000 instrument. The sample was degassed under vacuum for 6 h at 200 °C. The specific surface area was determined from the Brunauer, Emmett and Teller (BET) method. The Dubinin–Astakhov and Barrett–Joyner–Halenda methods were taken to determine the average pore sizes of microporous and mesoporous materials, respectively. Thermogravimetric analysis (TGA) was performed using a differential scanning calorimeter/thermogravimetric analyzer (Mettler-Toledo, TGA/DSC 1, Columbus, OH, USA) from 30 °C to 800 °C using 5 °C min in air atmosphere.

## 3. Results and Discussions

### 3.1. Effect of One-Step and Two-Step Catalyst on Preparation of Organo-Silica Xerogel

The FTIR spectra of one-step catalyzed (citric acid) and two-step catalyzed (acid-base) xerogels calcined at 175 °C in air is presented in Figure 2. All the xerogels, independently of their addition of step catalyst, showed similar vibrational bands in the region of 1400–600 cm^−1^. The peak of interest appearing at 940 cm^−1^ is attributed to the vibrational stretching of the silanol (Si-OH) groups. The other intense peak near 1070 cm^−1^ along with bands of 1170 and 1060 cm^−1^ were all assigned to various stretching and bending vibrations of the siloxane (Si-O-Si) groups. Meanwhile, the peak at 800 cm^−1^ was appropriated to the silica-carbon (Si-C) vibration band. A scan of the spectral profiles looks identical to previous study reported by Rahman et al. [25], which indicated at wavelength 958 cm^−1^ and 1280–760 cm^−1^ of silanol and siloxane, respectively. This is because the chemical constituents are tremendously similar in all the silica-carbon-based xerogels. The alteration of the vibrational bands connected to the silanol and siloxane concentration was quantitatively assessed by a deconvolution of the IR spectra bands at 940, 1060 and 800 cm^−1^.

The peak area ratio analysis regarding the silanol against the siloxane groups of one-step catalyst (citric acid) and two-step catalyst (acid-base) xerogels is displayed in Figure 3. The results exhibit that this ratio increases as the pH is increased from using one-step catalyst down to two-step catalyst. This behavior could be explained on the basis that the pH-dependency of the hydrolysis, condensation and polymerization reactions for porous properties of TEOS-derived silica have been reported extensively in several works [26,27,28]. These results clearly indicate that the lowest silanol concentrations, and likewise, the highest siloxane bridge concentrations, were achieved with calcined xerogels prepared with pH 6 (two-step catalyst) and pH 4.4 (one-step catalyst) (Figure 3).

In the sol–gel process by two-step catalysed (acid-base) xenogels, the first step was executed at pH ~4 (under an acidic condition) under reflux. Acid catalysed hydrolysis with heating promoted a high production of silanol species from the silane precursor of TEOS [29]. In the second step, the sol pH is adjusted by the addition of the ammonia hydroxide; the pH increases rapidly to >4, which is much higher than the isoelectric point boundary (pH 1–3) of the silica species. Instantly, the silanol species are expected to all be deprotonated while participating in the polycondensation reaction, generating a large concentration of highly siloxane species [9]. However, the result is contrary to Elma, Riskawati and Marhamah’s [9] work, which produced the highest silanol on two-step catalyst, as shown in Table 2. This suggests the difference of acid catalyst usage, e.g., organo catalyst (citric acid) instead of nitric acid in this work.

The citric acid acts as a catalyst and carbon source for tailoring organo-silica membrane, of which the structure and surface properties become stronger and have a good hydro-stability [2]. Despite this, the silanol species is generated the most in two-step catalyst compared to one-step catalyst, but the siloxane also forms the most in two-step catalyst (Table 2). The siloxane formation is promoted by condensation reactions during the addition of base catalyst. Moreover, the organo catalyst (e.g., citric acid) also favoured the siloxane formation, in line with results reported by Sumardi, Elma, Rampun, Lestari, Assyaifi, Darmawan, Yanto, Syauqiah, Mawaddah and Wati [20] in the designing of mesoporous hybrid organo-silica using organo-catalyst with different silica precursors (TEVS). It may be assumed that the Si-OH bonds were reduced and converted to be other bonds such as Si-C, C≡C, etc. This is due to the carbon content of citric acid as a catalyst [2]. In addition, the production of high siloxane groups in silica-based membrane has been reported to enhance the hydro-stability [16,17,30,31]. Therefore, based on the results, the presence of organo catalyst could produce more siloxane bridges and could decrease the hydro-instability.

Isotherms for the bulk organo-silica xerogels are shown in Figure 4 for both one- and two-step catalyst bulk xerogel samples. The isotherms for multiple xerogels are of type I, classified as microporous materials. Table 3 sorts the calculated values for the surface area (SBET), total pore volume, and average pore size. It is observed that one-step catalyst with the addition of citric acid may lead to an increase in both the SBET and total pore volume of 264 m^2^ g^−1^ and 0.0125 cm^3^ g^−1^, while the average pore size was slightly more reduced than the two-step catalyst. These results indicated that organo catalyst provided important qualitative information about the microstructure of the resulting molecular sieving membrane. The carbon content in xerogels during the calcination step remained in the matrices of silica and is expected to produce microporous materials, which potentially be the result of shrinkage of the silica framework during calcination of organo infiltrated into the silica pores [12,32].

The isotherm profiles of all the xerogels prepared at different step catalysis perform a quite similar nature. Xerogels synthesized in both one- and two-step catalyst show a tendency to form micro- and mesoporous material as the adsorption saturation is achieved above 0.65 P/P_0_ with capillary condensation leading to hysteresis approaching 0.04 P/P_0_. These results, verified with a higher number of siloxane species, are shown in Figure 2. On the other hand, type I isotherms with hysteresis indicated a typical micro/mesoporous material.

Although from plot N_2_, sorption isotherms in Figure 4 explained that the pore size is microporous material. All the organo-silica xerogels were categorized as slightly mesoporous structures, which have an average pore size range between 2 and 50 nm. It is shown that the pore size of both samples are 2.9 and 2.6 nm of two- and one-step catalyst samples, respectively (Table 3). Hence, meso-porosity correlates well with controlled concentrations of silanol/siloxane groups and appropriates with the deconvolution of the area ratio [Si-OH]/[Si-O-Si], as shown in Figure 3. These results are also in line with previous reports on silica-based membranes [26,28,33].

The xerogel used in this study was carbonized at 175 °C; the TGA combustion was carried out up to 800 °C to investigate the weight loss for all components of silica-carbon matrices. As seen in Figure 5, three regions can be explained about the weight loss of component masses. The first area is the loss of volatile components such as moisture, solvents, and monomers. The mass loss was decreased rapidly by ±10% at temperature 30–120 °C; this could be due to physiosorbed water removal in both one- and two-step catalyst samples. It normally happened because the silica material has a hydrophilic nature and easily absorbs water molecules through hydrogen bond (OH) groups of silanol species [34,35]. Hence, it could be remaining in the carbon-silica matrix, as observed in the IR spectrum in Figure 2. Based on the results, organo-silica xerogels prepared in one- and two-step using organo catalyst appear to be very similar to carbon-silica-based xerogel, which carbonized with surfactant, as reported by Duke et al. [36].

Figure 5 exhibits the xerogel prepared in two-step catalyst may absorb more water (15%) than the xerogel in one-step catalyst or organo catalyst (12%). The TG profile present at stage 2 experienced a decomposition process. The material starts to burn and decompose during this stage at 70 to 200 °C. All component materials are not completely decomposed in the second stage. Hereafter, the third stage occurred at temperatures up to 200 °C; the xerogel in two-step catalyst performed a large mass loss compared to one-step catalysts. It is clearly seen that without base addition, the sample is easier to decompose. The third stage phenomenon could have also been caused by carbon decomposition and combustion reactions. Figure 5 shows that the trends for all samples are similar to create flat lines. This means that the high temperature of calcination lead itself to the material completely decomposing. In addition, carbon from the citric acid chain easily burns at temperatures above 175 °C.

### 3.2. Effect of Sol pH in Organo-Silica Xerogels toward Structure and Fuctionalization Properties

The representative FTIR spectra for the organo-silica xerogels in different citric acid concentrations are presented in Figure 6a. The multiple citric acid concentration in silica sol were indicated by the sol pH of pH 4 (10 × 10^−2^ M C_6_H_8_O_7_), pH 6 (0.1 × 10^−2^ M C_6_H_8_O_7_) and pH 6.5 (0.07 × 10^−2^ M C_6_H_8_O_7_). The FTIR spectra shows the vibration band of silica and carbon compound at a wavelength region range of 1400–700 cm^−1^. Figure 6a displays that siloxane bridges (Si-O-Si) had stretching modes at bands near 1180, 1088 and 795 cm^−1^ for all varied pH, whilst, assigned at shoulder band, is silanol groups (Si-OH) at 940 cm^−1^. This result was similar to previous work reported by Rahman, Maimunawaro, Rahma, Isna and Elma [25] and Ayu Lestari et al. [37]; neither the functionalizing group of silanol or siloxane appeared in the samples. This is due to the xerogels that were fabricated containing silica and carbon from a templating agent (triblock copolymer P123). The silanol and siloxane vibration bands come forth because of the sol–gel process through hydrolysis and condensation reaction [13,38,39]. Silanol groups were formed during hydrolysis reaction and bridges formed under the condensation reaction. Other than that, deconvolution of FTIR data results by Fityk software is shown in Figure 6b.

The spectra FTIR data have been processed to peak fitting using Gaussian functional with error limit ±0.5% and using unit Q^n^ according to Park [40] publication. Based on Figure 6b, the result shows the highest relative peak area ratio of silanol/siloxane conducted at sol pH 6 of 0.3 Q^n^ (0.1 × 10^−2^ M C_6_H_8_O_7_). This result is higher than Elma et al.’s peak ratio, which only found 0.004–0.016 Q^n^ [41]. It might be that this work is preferred using an organic catalyst (citric acid) and other reports using inorganic catalyst (nitric acid) to prepare silica xerogel. Figure 6b shows that the peak area ratio increases by increasing the pH to pH 6, and dropped significantly at pH 6.5. This is due to the dependence of hydrolysis, condensation and polymerization reactions for the TEOS system described as the principle of the sol–gel process [42]. At pH 6.5, the ostwald maturation occurs, wherein in this condition there is a very rapid growth of particles under conditions and the weak polymer crosslinking formed at the first hydrolysis stage tends to weaken, even releasing and then settling. This is because the ratio of silanol/siloxane decreases at pH values above 6.

N_2_ isotherm curves of various pH are shown in Figure 7. As shown in Figure 7, both pH 6 and pH 6.5 have structure type IV isotherm with H4 hysteresis loops. Generally, the hysteresis loop H4 was a classification for the sample composed of ordered structures such as silica; meanwhile, type IV indicated mesoporous materials [42]. Xerogel at pH 6 was hysteresis from ~0.1 P/P_0_ to ~0.7 P/P_0_. These results exhibit a different structure because they simply used two-step catalyst (citric acid-ammonia), which was distinctive to Elma, Fitriani, Rakhman and Hidayati’s [15] work with two-step catalyst (nitric acid-ammonia). Moreover, calcination techniques have also contributed to the xerogel structure produced. The RTP (rapid thermal process) technique makes the xerogels become dense over the CTP (conventional thermal process) technique, which is the way the structure becomes micro/mesoporous. All samples show a similar trend by relative pressure measurement, even though differently from the pore volume, as shown in Table 4.

Table 4 displays the summary of surface properties of organo silica xerogels at varied citric acid concentrations, which represent the pH value of organo-silica sol. The S_BET_ of sol pH 6 calcined at 175 °C in air condition appears smaller than other types of xerogels in Table 4 of 234 m^2^ g^−1^. Excellently, pH 6.5 shows a higher surface area than other samples of 546 m^2^ g^−1^. Meanwhile, S_BET_ organo-silica sol pH 6.5 was higher than pH 6, at about 57%. This is because the high amount of organo catalyst concentration was added into the sol. The S_BET_ of organo-silica pH 6.5 in this work is higher by 4% and 18% than the carbon-silica and cobalt-oxide-silica xerogels [17,42], respectively. On the other hand, the total pore volume of organo-silica was also slightly higher by 26% over cobalt oxide silica xerogel like shown in Table 4. Moreover, all xerogels resulted in average pore sizes 2.05 and 2.21 nm of pH 6 and pH 6.5, respectively. It is concluded that the sols are potentially applied as a membrane for desalination application.

The TG analysis result was carried out to understand the mass loss behavior of a sample with varied citric acid concentrations (sol pH), as shown in Figure 8. It is interesting to observe that the mass loss profiles of organo-silica xerogels of different pH show a similar trend. The initial mass losses up to 100 °C were very similar and were mainly attributed to the desorption of water in the porous structure of both pH 6 and pH 6.5 organo-silica xerogel samples. After 250 °C, mass losses became significant as 17 wt% took place prior reaching ~280 °C, though these samples remained black, much akin to the color of carbon in the pH 6.5 sample. In both sample pH 6 and pH 6.5, the minor mass losses occurring in stage 2 at ~200 °C were mainly associated with further condensation reactions of the silanol group. These results suggest that the minor mass loss occurred due to the breaking down and/or loss of organic carbons groups in the organo catalyst. The mass loss curves in this work were different with the carbon-silica xerogel calcined at vacuum and N_2_ reported by Yang et al. [17]. A steady loss of the volatile organic occurred between 100 and 500 °C, which was caused by degradation of carbon from citric acid into smaller fragments continuously over this temperature.

## 4. Conclusions

The mesoporous organo-silica xerogel has been successfully produced using organo-acid via one-step and two-step (acid-base) catalyst by the sol–gel method. The results exhibit both the one-step and two-step catalyst obtaining good structure properties included high surface BET, pore volume and micro/mesoporous structure material. Nevertheless, two-step catalyst demonstrates that the functionalization has higher siloxane groups compared to the one-step catalyst, which is associated with good hydro-stability properties in those materials. It is also followed by the larger pore size in two-step catalyst that could be suitable for desalination application. In addition, the effect of pH on organo-silica xerogels toward psychochemical properties were observed. The pH 6.5 organo-silica xerogel by two-step catalyst showed excellent structure and functionalization properties that had high S_BET_, pore volume, mesoporous structure (2.2 nm) and a lower peak area ratio of silanol/siloxane groups. In the other hand, all organo-silica xerogels calcined at 175 °C were able to maintain the carbon bonds contained from organo acid in silica matrices. It is evidenced with TG analysis that up to 175 °C, the carbon decomposed.

## Figures and Tables

**Figure 1 membranes-11-00607-f001:**
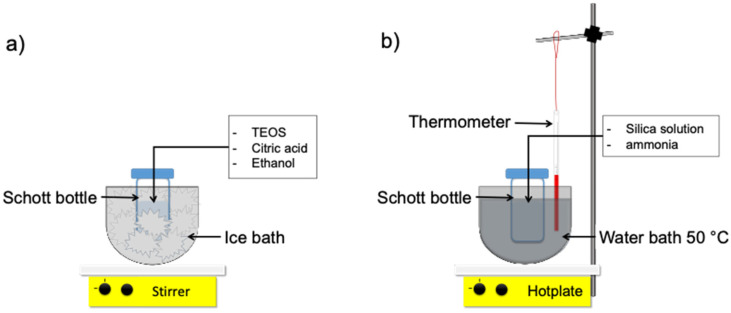
Schematic of organo-silica sol–gel process set-up (**a**) mixing process at 0 °C (**b**) mixing process at 50 °C.

**Figure 2 membranes-11-00607-f002:**
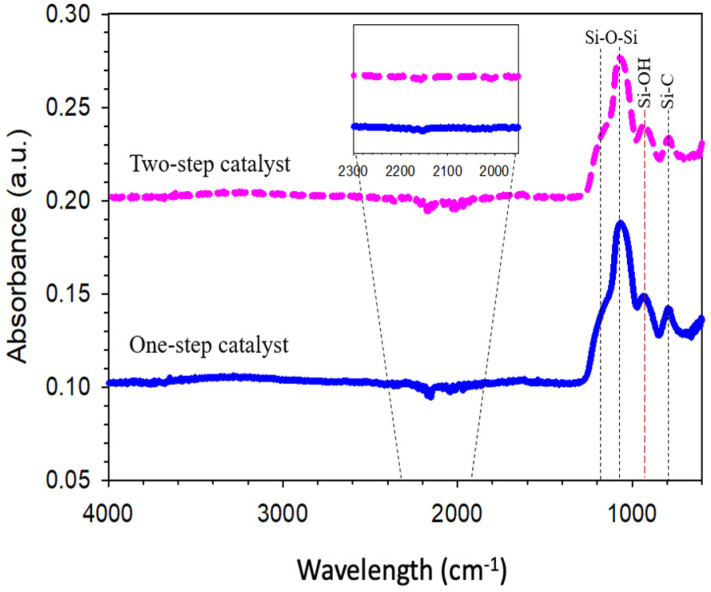
FTIR spectra of one-step catalyst (citric acid) and two-step catalyst (acid-base) xerogels calcined at 175 °C.

**Figure 3 membranes-11-00607-f003:**
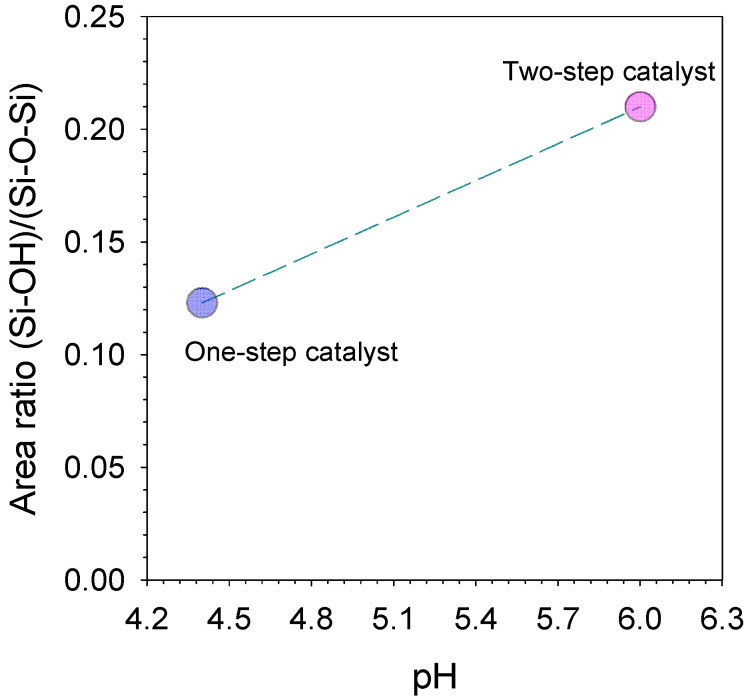
Deconvoluted peak area ratios of silanol/siloxane (940 cm^−1^)/(1070 cm^−1^) as a function of pH to perform various one-step catalyst (citric acid) and two-step catalyst (acid-base) xerogels.

**Figure 4 membranes-11-00607-f004:**
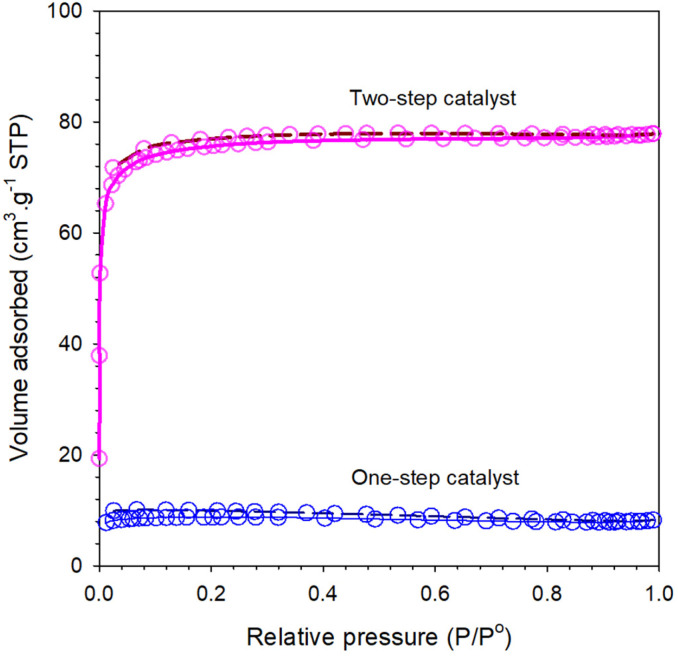
Plots of N_2_ isotherm data of the one-step catalyst and two-step catalyst xerogels calcined at 175 °C.

**Figure 5 membranes-11-00607-f005:**
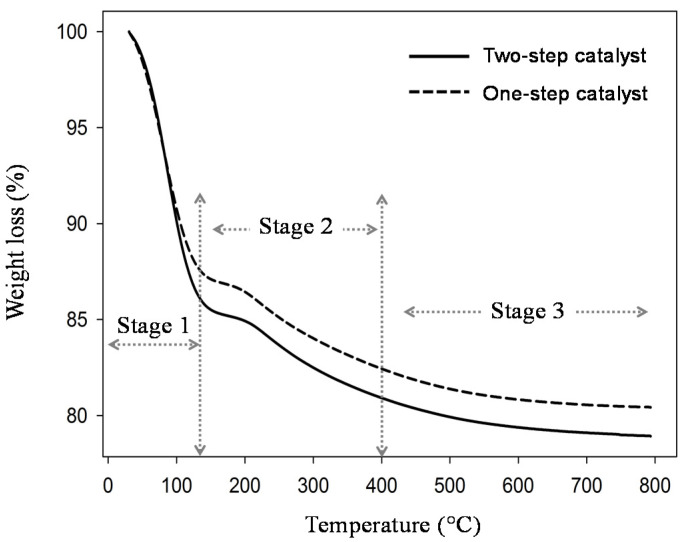
Weight loss curves of silica xerogel for dual catalyst (with ammonia) and single catalyst (without ammonia) as a function of temperature.

**Figure 6 membranes-11-00607-f006:**
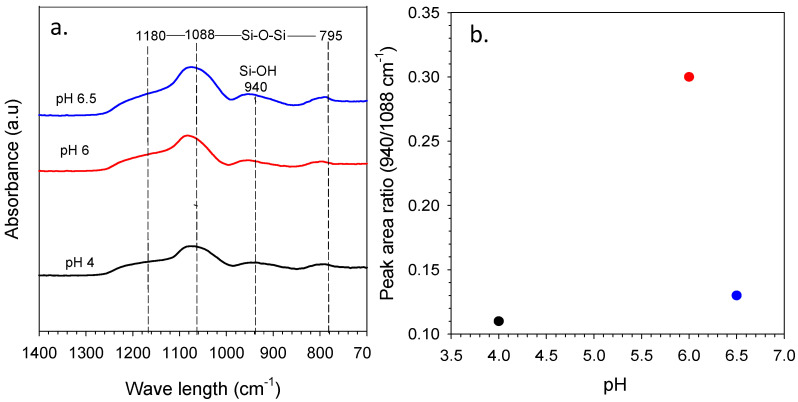
(**a**) FTIR spectra of organo-silica xerogels at varied pH and (**b**) the peak area ratio of silanol/siloxane for the function of sol pH.

**Figure 7 membranes-11-00607-f007:**
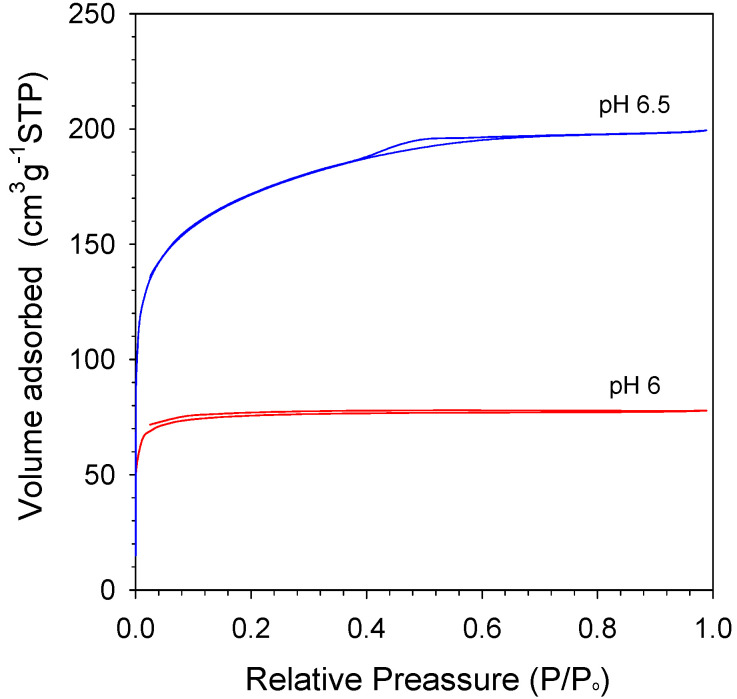
N_2_ physisorption isotherms of organo-silica xerogels at varied pH.

**Figure 8 membranes-11-00607-f008:**
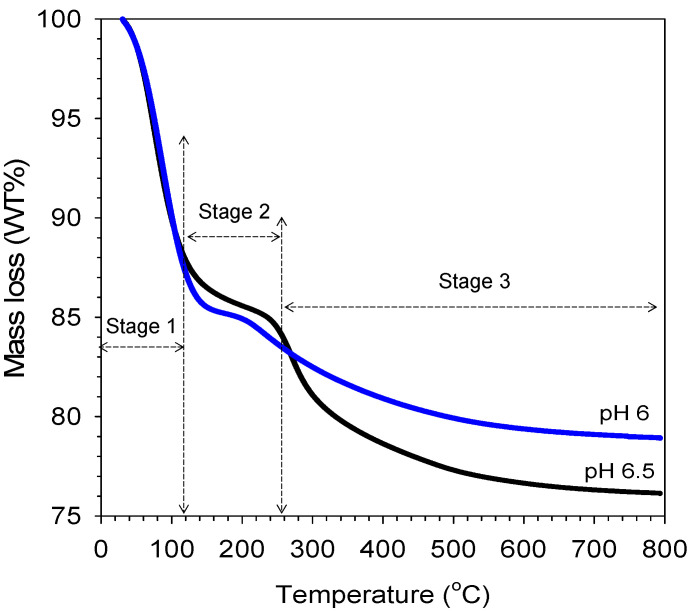
TGA mass loss curves of organo-silica xerogel in varied pH as function of temperature exposed.

**Table 1 membranes-11-00607-t001:** List of final molar ratio of organo-silica sol.

Sols	TEOS	EtOH [×10^1^]	C_6_H_8_O_7_ [×10^−2^]	NH_3_ [10^−3^]	H_2_O
One-step catalyst [pH 4.4]	1	3.8	0.1	-	5
Two-step catalyst [pH 4.4]	1	3.8	0.1	3	5
pH 4	1	3.8	10	3	5
pH 6	1	3.8	0.1	3	5
pH 6.5	1	3.8	0.07	3	5

**Table 2 membranes-11-00607-t002:** Deconvolution of Si-OH/Si-O-Si concentration.

Xerogels	Sol pH	Area (Q^n^)			Area Ratio Si-OH/Si-O-Si
		Si-O-Si	Si-OH	Si-C	
Two-step catalyst	6	4.999	1.039	0.339	0.207
One-step catalyst	4.4	4.478	0.540	1.036	0.120

**Table 3 membranes-11-00607-t003:** Surface properties of the organo-silica xerogels.

Xerogel	pH	S_BET_ (m^2^ g^−1^)	Pore Volume (cm^3^ g^−1^)	Average Pore Diameter (nm)
Two-step catalyst	6	234.273	0.012015	2.9208
One-step catalyst	4.4	264.276	0.01251	2.5939

**Table 4 membranes-11-00607-t004:** Surface properties of organo-silica xerogels at varied types.

Xerogels Types	Materials/Catalyst	Calcined Temp. (°C)	S_BET_ (m^2^ g^−1^)	Pore Volume (cm^3^ g^−1^)	Average Pore Diameter (nm)	Ref.
Organo-silica pH 6 (calcined in air)	TEOS/citric acid-ammonia	175	234	0.12	2.05	This work
Organo-silica pH 6.5 (calcined in air)	TEOS/citric acid-ammonia	175	546	0.31	2.21	This work
Carbon-silica (calcined in vacuum)	TEVS-P123/nitric acid-ammonia	450	761	0.62	2	[12]
Carbon-silica (calcined in N_2_)	TEVS-P123/nitric acid-ammonia	450	526	0.34	2.56	[17]
Cobalt oxide silica (calcined in vacuum)	TEOS-cobalt/ammonia	600	450	0.23	<2	[42]
Cobalt oxide silica (calcined in vacuum)	ES40-cobalt/ammonia	600	440	0.18	>2	[42]

## Data Availability

Not applicable.

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
