# Peer review of "Physicochemical Properties of Mesoporous Organo-Silica Xerogels Fabricated through Organo Catalyst"

_membranes, 2021, doi:10.3390/membranes11080607_

Round 1
Reviewer 1 Report
This work is scientifically well fundamented, the tests are in accordance to the characterization of the obtained materials and the explanations are pertinent, but the presentation is poor. As mentioned at the bottom of my observations, English language must thoroughly be revised, in order to clarify the paper.
line 33: which effectively should be replaced with: which are effective
line 39: concern to developed should be replaced with: concern to development; espacially should be replaced with especially
line 42: inorganic based materials offering should be replaced with inorganic based materials are offering
line 44: a good chemically stable and affordable to employed should be replaced with a good chemical stability and is affordable to be employed
line 50 carbon sources should be replaced with carbon source
line 51: compare to should be replaced with compared to
line 60: thus is to be replaced with these
line 63: one of mineral material should be replaced with one of the mineral materials; in oerder should be replaced with in order
line 65: tailor pore should be replaced with tailored pore
line 66: process compound through should be replaced with process composed of
lines 67-69: the entire phrase should be reformulated. For example: The high viscosity has the effect of changing the suspense colloid phase (sol) into continuous liquid phase
line 69: the entire phrase should be reformulated. For example: The thin film as product has the following characteristics: it is a transparent liquid and it has a high viscosity
line 72: was depended on should be replaced with was depending on.
Decreasing of pH silica sol was affected to form the pore size of silica matrices to become smaller (microporous) should be reformulated. For example: The decrease of the pH in the silica sol determines the shrinking of the pore size in the silica matrices to micropores
line 74: However, the desalination process is better to apply the bigger particular pore sizes in order to separate contaminant and water molecular could through membrane – should be completely revised. For example: However, for the desalination process it is better to apply the bigger particular pore sizes in order to separate contaminants and the water molecules could pass through the membrane
line 78: based catalysts should be replaced with base catalysts
line 79: There are variety precursors should be replaced with there is a variety of precursors
line 83: a polymerization processes should be replaced with a polymerization process
line 84: The advantages of silica thin films with is the capability to create a stronger matrices should be reformulated
line 85: this material lack should be replaced with this material lacks, without of
line 86: P123 strengthen the matrix should be replaced with P123 strengthens the matrix
line 90: may a good option should be replaced with may be a good option
line 91: The advantage of presence carbon compound elements in silica matrices are able to improve strengthen of the bonds in the matrix- should be reformulated. For example: The advantage of the presence of carbon compound elements in silica matrices is that they are able to improve and strengthen the bonds in the matrix
line 93: that is easy should be replaced with that are easy
line 93-94: rephrase: In this study an inorganic base catalyst, namely ammonia was used
line 98: compare should be replaced with compared
line 103: correct: materials and chemicals that have been employed
line 106: demineralized water
line 111: correct: was synthesized by the simple method
line114: demineralized
line 116: correct: like shown
line 117: correct: then 20 mL EtOH were added under stirring
line 120: correct: Then ammonia was drop-wise added
line 121: correct: The final pH of the silica sols
line 126: correct: xerogel was ground
line 146: consider revising: Characterization results of ??? by FTIR spectra shows (Is presented) in Figure 2.
line 149: consider revising: Whilst assign at shoulder band is groups (Si-OH) at 940 cm-1.
line 155: correct: silanol groups were formed
line 162: consider revising: It suggested due to in this work is used different acid catalyst of organic acid instead inorganic acid
line 173: consider revising: The highest area of ??? and highest ??? is obtained at pH 4 and pH 6.5
line 178: consider revising: This result differed from dual catalyst sol gel, which delivers structure???
line 185: consider revising: Table 1 shows summary of BET surface properties of ??? at varied types ???
The English language is very poor, an extensive revision of the entire paper is required, for an easier read of the work. Examples are plenty given above
line 332: employing sol gel process through organo and base catalysts.: revised: employing sol gel process through organo-acid or organo/citric acid and base catalysts
Author Response
Response to Reviewer Comments
Please refer to the file attached
Reviewer #1:
This work is scientifically well fundamented, the tests are in accordance to the characterization of the obtained materials and the explanations are pertinent, but the presentation is poor. As mentioned at the bottom of my observations, English language must thoroughly be revised, in order to clarify the paper.
Comment 1:
line 33: which effectively should be replaced with: which are effective
line 39: concern to developed should be replaced with: concern to development; especially should be replaced with especially
line 42: inorganic based materials offering should be replaced with inorganic based materials are offering
line 44: a good chemically stable and affordable to employed should be replaced with a good chemical stability and is affordable to be employed
line 50 carbon sources should be replaced with carbon source
line 51: compare to should be replaced with compared to
line 60: thus is to be replaced with these
line 63: one of mineral material should be replaced with one of the mineral materials; in oerder should be replaced with in order
line 65: tailor pore should be replaced with tailored pore
line 66: process compound through should be replaced with process composed of
lines 67-69: the entire phrase should be reformulated. For example: The high viscosity has the effect of changing the suspense colloid phase (sol) into continuous liquid phase
line 69: the entire phrase should be reformulated. For example: The thin film as product has the following characteristics: it is a transparent liquid and it has a high viscosity
line 72: was depended on should be replaced with was depending on.
Decreasing of pH silica sol was affected to form the pore size of silica matrices to become smaller (microporous) should be reformulated. For example: The decrease of the pH in the silica sol determines the shrinking of the pore size in the silica matrices to micropores
line 74: However, the desalination process is better to apply the bigger particular pore sizes in order to separate contaminant and water molecular could through membrane – should be completely revised. For example: However, for the desalination process it is better to apply the bigger particular pore sizes in order to separate contaminants and the water molecules could
pass through the membrane
line 78: based catalysts should be replaced with base catalysts
line 79: There are variety precursors should be replaced with there is a variety of precursors
line 83: a polymerization processes should be replaced with a polymerization process
line 84: The advantages of silica thin films with is the capability to create a stronger matrices should be reformulated
line 85: this material lack should be replaced with this material lacks, without of
line 86: P123 strengthen the matrix should be replaced with P123 strengthens the matrix
line 90: may a good option should be replaced with may be a good option
line 91: The advantage of presence carbon compound elements in silica matrices are able to improve strengthen of the bonds in the matrix- should be reformulated. For example: The advantage of the presence of carbon compound elements in silica matrices is that they are able to improve and strengthen the bonds in the matrix
line 93: that is easy should be replaced with that are easy
line 93-94: rephrase: In this study an inorganic base catalyst, namely ammonia was used
line 98: compare should be replaced with compared
line 103: correct: materials and chemicals that have been employed
line 106: demineralized water
line 111: correct: was synthesized by the simple method
line114: demineralized
line 116: correct: like shown
line 117: correct: then 20 mL EtOH were added under stirring
line 120: correct: Then ammonia was drop-wise added
line 121: correct: The final pH of the silica sols
line 126: correct: xerogel was ground
line 146: consider revising: Characterization results of ??? by FTIR spectra shows (Is presented) in Figure 2.
line 149: consider revising: Whilst assign at shoulder band is groups (Si-OH) at 940 cm-1.
line 155: correct: silanol groups were formed
line 162: consider revising: It suggested due to in this work is used different acid catalyst of organic acid instead inorganic acid
line 173: consider revising: The highest area of ??? and highest ??? is obtained at pH 4 and pH 6.5
line 178: consider revising: This result differed from dual catalyst sol gel, which delivers structure???
line 185: consider revising: Table 1 shows summary of BET surface properties of ??? at varied types ???
The English language is very poor, an extensive revision of the entire paper is required, for an easier read of the work. Examples are plenty given above
line 332: employing sol gel process through organo and base catalysts.: revised: employing sol gel process through organo-acid or organo/citric acid and base catalysts
Response 1: To reviewer, thank you very much for the review, correction, suggestion, and, we are very happy to Reviewer 1 for the advice and corrections.
we have added some paragraph as well in order to improves our manuscript context into the manuscript especially in the introduction section.

Reviewer 2 Report
The presented paper is devoted to the study on the physicochemical properties of organo-silica xerogels derived from organocatalyst, including the effect of the addition of citric acid and ammonia as dual catalysts. The organo-silica xerogels were characterized by FTIR, TGA, and nitrogen sorption.
I suppose that the paper is not suitable for publication in the present form due to several reasons:
- The introduction is poorly structured. The novelty of the work is not clear. How the article material differs from the publication of the authors [Membrane Technology, Volume 2021, Issue 2, 2021, P. 5-8, https://doi.org/10.1016/S0958-2118(21)00029-X], the reference to which is not given?
- Table 2 – unacceptable terms are given.
- The work contains 35 self-citations among 57 references, which is unacceptable. References are designed carelessly in the text and in the final list.
- Extensive English editing is needed because the manuscript contains many typos and grammatical errors. The article is not readable and requires serious revision.
Author Response
Response to Reviewer Comments
Reviewer #2:
The presented paper is devoted to the study on the physicochemical properties of organo-silica xerogels derived from organocatalyst, including the effect of the addition of citric acid and ammonia as dual catalysts. The organo-silica xerogels were characterized by FTIR, TGA, and nitrogen sorption.
I suppose that the paper is not suitable for publication in the present form due to several reasons:
Comment 1:
The introduction is poorly structured. The novelty of the work is not clear. How the article material differs from the publication of the authors [Membrane Technology, Volume 2021, Issue 2, 2021, P. 5-8, https://doi.org/10.1016/S0958-2118(21)00029-X], the reference to which is not given?
Response 1: To reviewer, thank you very much for the review, correction, suggestion, and, we are happy to Reviewer for advice, we add some paragraph to clear the novelty into manuscript in introduction section.
“In this study, the addition of citric acid catalyst may be a good option. This citric acid (C6H8O7) containing carbon from glucose could be used as an organic catalyst [19]. The advantage of the presence of carbon compound elements in silica matrices is that they able to improve and strengthen the bonds in the matrix [20]. Also, it originally comes from organic acids that are easy to find and cheap. In this study, an inorganic base catalyst, namely ammonia was used. It is to increase the pH of thin films so that pH from thin films is not too acidic. Because of the pH is too acidic it will produce microporous pore size which is not suitable to be applied to water desalination.”
Comment 2:
Table 2 – unacceptable terms are given.
Response 2: thank you to reviewer for pointing out the miss writing in our manuscript. We have been revised the terms into the manuscript.
Comment 3:
The work contains 35 self-citations among 57 references, which is unacceptable. References are designed carelessly in the text and in the final list.
Response 3: We apologise about that; we have been reduced and rearrangement the reference to be not too many self-citation in the manuscript
Comment 4:
Extensive English editing is needed because the manuscript contains many typos and grammatical errors. The article is not readable and requires serious revision.
Response 4: Thank you for the advice. We are trying our best to avoid grammatical mistakes. We regret there were problems with the English. This manuscript has been carefully revised in order to improve the grammatical errors as well as typos. We have also noted revisions and modified the manuscript that have been corrected.

Reviewer 3 Report
In the article of A. Sumari it was presented the studies of the physicochemical properties of organo-silica xerogels derived from organo catalyst. The publication shows a known method of preparing xerogels using a mixture of catalysts (citric acid / NH3) and testing the obtained membranes. The membranes were tested for N2 sorption. The scope of the presented research does not affect the development of knowledge in the field of xserogel membranes and has already been the subject of other described works, e.g. https://iopscience.iop.org/article/10.1088/1755-1315/175/1/012008/pdf, https://www.e3s-conferences.org/articles/e3sconf/pdf/2020/08/e3sconf_etmc2020_07008.pdf. The manuscript describes the results based on earlier work. This is evidenced by the large number of auto cited works covering over half of the references. In conclusion, I do not recommend it this manuscript for publication.
Author Response
Response to Reviewer Comments
Please refer to the file attached
Reviewer #3:
In the article of A. Sumari it was presented the studies of the physicochemical properties of organo-silica xerogels derived from organo catalyst. The publication shows a known method of preparing xerogels using a mixture of catalysts (citric acid / NH3) and testing the obtained membranes. The membranes were tested for N2 sorption. The scope of the presented research does not affect the development of knowledge in the field of xerogel membranes and has already been the subject of other described works, e.g.
https://iopscience.iop.org/article/10.1088/1755-1315/175/1/012008/pdf, https://www.e3sconferences.org/articles/e3sconf/pdf/2020/08/e3sconf_etmc2020_07008.pdf. The manuscript describes the results based on earlier work. This is evidenced by the large number of auto cited works covering over half of the references. In conclusion, I do not recommend it this manuscript for publication.
Response 4: To reviewer, thank you very much for the review and correction. This manuscript has different discourse with our previous publication especially in pore structure. The citric acid concentration was varied in this work, moreover, the utilized of one step and two step catalyst were comparing to this work.

Round 2
Reviewer 1 Report
The conclusions should be better rephrased, in order to express more clearly the results of the paper. The work was improved, it is more clear, still there are some issues with the language, most of which I present below:
line 45: the utilize should be replaced with THE USE
Lines 62-63: As a consequence, the matrix leading to dissolution and/or densification. should be rephrased
line 64: Ineffectually is an adverb and does not refer to any verb; should be reconsidered (maybe inefficiency?)
line 89: a based catalyst should be replaced with a base catalyst
line 97: they able should be replaced with they are able
line 121: TEOS has added replace with; TEOS is added, was added…
line 154: assign should be replaced with assigned
line 174: In another word should be replaced with in other words
line 174: that the ratio increased by increasing pH, however, at pH 6.5 is decreased should be rephrased. It does not indicate what kind of ratio is referred to
line 187: The RTP (rapid thermal process) technique bring out the xerogels become dense over the CTP (conventional thermal process) technique like Elma, et al. work on vacuum condition should be rephrased
line 198: Meanwhile, pH 6.5 in this work was much high if compared to silica pH 6 and pH 7 should be reconsidered
line 201: Carbon compound in the silica thin film was provided more surface area than without it. However, pH 6.5 is lower compared to P123 [34]. It is caused P123 have long carbon chains over citric acid so that offering a higher surface area - should be rephrased, it is not clear
line 224: to applicate should be replaced with to apply
line 228: shown should be replaced with show
line 253: It is shown the pore size are should be replaced it is shown that the pore size
line 257: shown should be replaced with shows
from line 266 there is a repetition, where the authors should be more careful
line 305: burn should be replaced with burned
line 314: Sol gel process was chosen as the method in synthesize mesoporous silica should be replaced with: The sol gel process was chosen as the method for the synthesis of mesoporous silica
line 315: The results showed that for thin-film using single or dual catalysts gives different characters should be rephrased
line 316: Where the single catalyst (organo/citric acid) gives a higher Si-O-Si and Si-C concentration than a thin film that is employed as a dual catalyst (base catalyst) should be rephrased
line 312: burn should be replaced with burned
Author Response
REVIEWER 1
Comment 1:
The conclusions should be better rephrased, in order to express more
clearly the results of the paper. The work was improved, it is more
clear, still there are some issues with the language, most of which I
present below:
line 45: the utilize should be replaced with THE USE
Lines 62-63: As a consequence, the matrix leading to dissolution and/or
densification. should be rephrased
line 64: Ineffectually is an adverb and does not refer to any verb;
should be reconsidered (maybe inefficiency?)
line 89: a based catalyst should be replaced with a base catalyst
line 97: they able should be replaced with they are able
line 121: TEOS has added replace with; TEOS is added, was added…
line 154: assign should be replaced with assigned
line 174: In another word should be replaced with in other words
line 174: that the ratio increased by increasing pH, however, at pH 6.5
is decreased should be rephrased. It does not indicate what kind of
ratio is referred to
line 187: The RTP (rapid thermal process) technique bring out the
xerogels become dense over the CTP (conventional thermal process)
technique like Elma, et al. work on vacuum condition should be rephrased
line 198: Meanwhile, pH 6.5 in this work was much high if compared to
silica pH 6 and pH 7 should be reconsidered
line 201: Carbon compound in the silica thin film was provided more
surface area than without it. However, pH 6.5 is lower compared to P123
[34]. It is caused P123 have long carbon chains over citric acid so that
offering a higher surface area - should be rephrased, it is not clear
line 224: to applicate should be replaced with to apply
line 228: shown should be replaced with show
line 253: It is shown the pore size are should be replaced it is shown
that the pore size
line 257: shown should be replaced with shows
from line 266 there is a repetition, where the authors should be more
careful
line 305: burn should be replaced with burned
line 314: Sol gel process was chosen as the method in synthesize
mesoporous silica should be replaced with: The sol gel process was
chosen as the method for the synthesis of mesoporous silica
line 315: The results showed that for thin-film using single or dual
catalysts gives different characters should be rephrased
line 316: Where the single catalyst (organo/citric acid) gives a higher
Si-O-Si and Si-C concentration than a thin film that is employed as a
dual catalyst (base catalyst) should be rephrased
line 312: burn should be replaced with burned
Response 1:
To reviewer, thank you very much for the review, advice correction as well as the suggestion.
We are happy to follow them. In the latest manuscript, we have tried our best to totally revised the manuscript and also rephrased sentences. We also have improved the explanation of our work results on the paper massively. We are terribly sorry about the language issue shown in the previous manuscript, however, we have carefully improved the English in every sentences so that this manuscript is much more better to deliver.
Again, thank you very much.

Reviewer 2 Report
Comment 1:
The introduction is poorly structured. The novelty of the work is not clear. How the article material differs from the publication of the authors [Membrane Technology, Volume 2021, Issue 2, 2021, P. 5-8, https://doi.org/10.1016/S0958-2118(21)00029-X], the reference to which is not given?
Response 1: To reviewer, thank you very much for the review, correction, suggestion, and, we are happy to Reviewer for advice, we add some paragraph to clear the novelty into manuscript in introduction section.
“In this study, the addition of citric acid catalyst may be a good option. This citric acid (C6H8O7) containing carbon from glucose could be used as an organic catalyst [19]. The advantage of the presence of carbon compound elements in silica matrices is that they able to improve and strengthen the bonds in the matrix [20]. Also, it originally comes from organic acids that are easy to find and cheap. In this study, an inorganic base catalyst, namely ammonia was used. It is to increase the pH of thin films so that pH from thin films is not too acidic. Because of the pH is too acidic it will produce microporous pore size which is not suitable to be applied to water desalination.”
Comment to response:
This answer does not provide information about the difference of the material presented in the paper from the material published earlier [Membrane Technology, Volume 2021, Issue 2, 2021, P. 5-8, https://doi.org/10.1016/S0958-2118(21)00029-X], the reference to which is not given. The novelty of the work is still unclear.
Comment 3:
The work contains 35 self-citations among 57 references, which is unacceptable. References are designed carelessly in the text and in the final list.
Response 3: We apologise about that; we have been reduced and rearrangement the reference to be not too many self-citation in the manuscript
Comment to response:
The presented version of the article also has problems with self-citation: more than 50% are references to their own works
Comment 4:
Extensive English editing is needed because the manuscript contains many typos and grammatical errors. The article is not readable and requires serious revision.
Response 4: Thank you for the advice. We are trying our best to avoid grammatical mistakes. We regret there were problems with the English. This manuscript has been carefully revised in order to improve the grammatical errors as well as typos. We have also noted revisions and modified the manuscript that have been corrected.
Comment to response:
Despite the attempts to correct the text, it is far from perfect and requires extensive English editing.
Author Response
REVIEWER 2
Comment 1:
The introduction is poorly structured. The novelty of the work is not clear. How the article material differs from the publication of the authors [Membrane Technology, Volume 2021, Issue 2, 2021, P. 5-8, https://doi.org/10.1016/S0958-2118(21)00029-X], the reference to which is not given?
Response 1:
To reviewer, thank you very much for the review, advice, correction and suggestion.
We are happy for that and we have d some paragraphs to make it clearer showing the novelty written in the manuscript especially in the introduction.
“In this study, the addition of citric acid catalyst may be a good option. This citric acid (C6H8O7) containing carbon from glucose could be used as an organic catalyst [19]. The advantage of the presence of carbon compound elements in silica matrices is that they able to improve and strengthen the bonds in the matrix [20]. Also, it originally comes from organic acids that are easy to find and cheap. In this study, an inorganic base catalyst, namely ammonia was used. It is to increase the pH of thin films so that pH from thin films is not too acidic. Because of the pH is too acidic it will produce microporous pore size which is not suitable to be applied to water desalination.”
Comment to response:
This answer does not provide information about the difference of the material presented in the paper from the material published earlier [Membrane Technology, Volume 2021, Issue 2, 2021, P. 5-8, https://doi.org/10.1016/S0958-2118(21)00029-X], the reference to which is not given. The novelty of the work is still unclear.
Response 2:
To reviewer, thank you very much for the review, correction, and suggestion. The difference between our work and the published article is the utilization of silane precursor. The previous work was employed the dual precursors of TEVS/TEOS, instead in this manuscript is only using single precursor of TEOS. To answer that concern, we already fully revised our introduction section to provide more information about the most recent of our work.
Comment 3:
The work contains 35 self-citations among 57 references, which is unacceptable. References are designed carelessly in the text and in the final list.
Comment to response:
The presented version of the article also has problems with self-citation: more than 50% are references to their own works
Response 3:
Dear Reviewer, we are terribly apologised about this issue.
We have removed the several self-citations in the manuscript that is far away related to our work.
Comment 4:
Extensive English editing is needed because the manuscript contains many typos and grammatical errors. The article is not readable and requires serious revision.
Response 4:
Thank you for the advice.
We have put our best to avoid grammatical mistakes for this manuscript. We regret there were problems with the English. This manuscript has been carefully revised in order to improve the grammatical errors as well as typos. We have also noted the revisions and we have modified the manuscript become much more better.
Comment to response:
Despite the attempts to correct the text, it is far from perfect and requires extensive English editing.
Response 4:
Dear reviewer, we are really sorry about the language issue written in the last manuscript.
We have carefully worked to improve the English in this manuscript, so that this manuscript is totally improved as well as we do avoid the grammatical errors and typos.
REVIEWER 3
Considering the journal's rank as well as the scope of the presented research (revised version), I recommend moving this manuscript to another journal, i.e. Membrane Technology. Nevertheless, I leave it to the editor's decision. My comments on the revised version are presented below.
According to the explanations provided, the authors claim that this work includes the effect of the concentration of citric acid as well as the comparison of a one-stage (acid) and two-stage (acid + base) catalyst on prepared mesoporous structures. Mainly, the change of these factors influences the size of the pores obtained (2.9 and 2.5 nm), so the obtained structures can be used for water desalination. Moreover, it was also found that mesoporous structures not significantly different in pore size were obtained by both approaches but with a different bond composition mainly covering: Si-OH and Si-C. In the revised version, there are still some elements that require explanation and correction:
- The synthesis is not fully described. Materials were obtained using acid (as a single catalyst) as well as acid and ammonia (double catalyst). Nevertheless, a description of the synthesis 2.2. It is vague and does not fully account for the synthetic assumptions of a single catalyst (three samples) and a double catalyst (one sample). It should be clear at this stage how many samples were prepared and how.
- The recipe states that the citric acid solutions used for the synthesis were obtained from 0.001 M citric acid solution. In the further part of the study, the normal concentrations of citric acid (0.1, 0.0001, 0.00007 N) were mentioned. In general, the using of normal concentration (especially for weak acids) is unclear. It would be more appropriate to give the molar concentration of the acid solutions used. Moreover, it is not clear how 0.1N of acid was obtained from 0.001M of acid solution? It should be mentioned that IUPAC does not recommend the use of this type of concentration determination.
-In the abstract it states that: “The final molar ratio of organo-silica was 1:38:x:y:5 (TEOS:ethanol: citric acid: NH3:H2O)”. However, in chapter 2.2. (Synthesis of sol-gel process) the starting materials (amounts) used to obtain the silica material are listed. 20 ml of EtOH (99%, g=0.789g/cm3, M=46.07g/mol, therefore it is 0.545 mol) and 18.66 gr TEOS (99%, M=208.33 g/mol, therefore it is 0.0887 mol) in this synthesis was used. It follows that TEOS to EtOH is in a molar ratio of approximately 1: 6, not 1:38. Thus, the molar ratio presented in the abstract has no coverage in the material preparation procedure.
- Moreover, in the sentence quoted above: “The final molar ratio of organo-silica was 1:38:x:y:5 (TEOS:ethanol: citric acid: NH3:H2O)”- indicates on the use of ammonia in a molar ratio of 0 to 0.0003. However, in the synthesis and in the further part of the work, there are no considerations about the amount of ammonia used. Only one example obtained with a double catalyst is described. So why is there y in the abstract, suggesting a different ammonia content?
- In the synthesis description, there is no information as to how it was added the second catalyst, i.e. ammonia. The only information is that it was added dropwise. What was its volume and whether it was diluted with water? It would be good to specify whether the dropwise addition took place in a short time (right away) or maybe within 30 minutes. These are important elements in the preparation of silica materials, so it would be good to supplement them.
- There is no information whether magnetic stirring was used in the second stage of the process - slow or fast mixing, or maybe no mixing was used?
- Sentence in chapter 3.1.: “The multiple citric acid concentration in silica sol were indicated by the pH of sol is pH 4 (0.1 N C6H8O7), pH 6 (0.0001 N C6H8O7) and pH 6.5 (0.007 C6H8O7)” – is inconsistent with the information in chapter 2.2 and the rest of the description in the manuscript.
- Sentence on page 6: “Moreover, all resulted in a structure with pore sizes 2.05 and 2.21 nm so that the sol could be applied as thin-film silica on the membrane. could be used in the process of desalination of water through pervaporation” contains error.
- based on the description on page 10: “The sample with a lower pH value gives a higher surface area (234.273 m²/g). Vice versa, the higher pH give higher surface area (264.276 m²/g)” - it is difficult to tell which pH is responsible for obtaining a higher surface area.
- As for the selection of literature, there are still citing post-conference materials instead indexed and significant scientific in this subject journals
Response :
Dear reviewer, thank you very much for the review, correction and suggestion.
We are happy for advice and we have tried our best to majorly revised the manuscript that briefly added some elements that require much more explanation and correction through added more the sentences and correction in the manuscript, as follow:
- The synthesis is already revised and added more sentences in order to complete the description of our work in the methods section.
- We feel guilty about so much mistakes found in our previous manuscript, however, we have tried our best to revise our manuscript to become much more better and improve the explanation about the molar ratio and other things needed.
- The molar ratio of our organo-silica sol preparation was clearly listed on the Table 1.
- The brief of method was also already added in the revised manuscript.
- Thank you for the reviewer, which can spend reviewer’s spare time for correcting the mistakes and give the advices for improving our manuscript. We have been revised our manuscript and tried our best to avoid the mistakes in our paper.
- We are terribly apologised about that; we have been removing the several self-citations in the manuscript.
Thank you very much

Reviewer 3 Report
Considering the journal's rank as well as the scope of the presented research (revised version), I recommend moving this manuscript to another journal, i.e. Membrane Technology. Nevertheless, I leave it to the editor's decision. My comments on the revised version are presented below.
According to the explanations provided, the authors claim that this work includes the effect of the concentration of citric acid as well as the comparison of a one-stage (acid) and two-stage (acid + base) catalyst on prepared mesoporous structures. Mainly, the change of these factors influences the size of the pores obtained (2.9 and 2.5 nm), so the obtained structures can be used for water desalination. Moreover, it was also found that mesoporous structures not significantly different in pore size were obtained by both approaches but with a different bond composition mainly covering: Si-OH and Si-C. In the revised version, there are still some elements that require explanation and correction:
- The synthesis is not fully described. Materials were obtained using acid (as a single catalyst) as well as acid and ammonia (double catalyst). Nevertheless, a description of the synthesis 2.2. It is vague and does not fully account for the synthetic assumptions of a single catalyst (three samples) and a double catalyst (one sample). It should be clear at this stage how many samples were prepared and how.
- The recipe states that the citric acid solutions used for the synthesis were obtained from 0.001 M citric acid solution. In the further part of the study, the normal concentrations of citric acid (0.1, 0.0001, 0.00007 N) were mentioned. In general, the using of normal concentration (especially for weak acids) is unclear. It would be more appropriate to give the molar concentration of the acid solutions used. Moreover, it is not clear how 0.1N of acid was obtained from 0.001M of acid solution? It should be mentioned that IUPAC does not recommend the use of this type of concentration determination.
-In the abstract it states that: “The final molar ratio of organo-silica was 1:38:x:y:5 (TEOS:ethanol: citric acid: NH3:H2O)”. However, in chapter 2.2. (Synthesis of sol-gel process) the starting materials (amounts) used to obtain the silica material are listed. 20 ml of EtOH (99%, g=0.789g/cm3, M=46.07g/mol, therefore it is 0.545 mol) and 18.66 gr TEOS (99%, M=208.33 g/mol, therefore it is 0.0887 mol) in this synthesis was used. It follows that TEOS to EtOH is in a molar ratio of approximately 1: 6, not 1:38. Thus, the molar ratio presented in the abstract has no coverage in the material preparation procedure.
- Moreover, in the sentence quoted above: “The final molar ratio of organo-silica was 1:38:x:y:5 (TEOS:ethanol: citric acid: NH3:H2O)”- indicates on the use of ammonia in a molar ratio of 0 to 0.0003. However, in the synthesis and in the further part of the work, there are no considerations about the amount of ammonia used. Only one example obtained with a double catalyst is described. So why is there y in the abstract, suggesting a different ammonia content?
- In the synthesis description, there is no information as to how it was added the second catalyst, i.e. ammonia. The only information is that it was added dropwise. What was its volume and whether it was diluted with water? It would be good to specify whether the dropwise addition took place in a short time (right away) or maybe within 30 minutes. These are important elements in the preparation of silica materials, so it would be good to supplement them.
- There is no information whether magnetic stirring was used in the second stage of the process - slow or fast mixing, or maybe no mixing was used?
- Sentence in chapter 3.1.: “The multiple citric acid concentration in silica sol were indicated by the pH of sol is pH 4 (0.1 N C6H8O7), pH 6 (0.0001 N C6H8O7) and pH 6.5 (0.007 C6H8O7)” – is inconsistent with the information in chapter 2.2 and the rest of the description in the manuscript.
- Sentence on page 6: “Moreover, all resulted in a structure with pore sizes 2.05 and 2.21 nm so that the sol could be applied as thin-film silica on the membrane. could be used in the process of desalination of water through pervaporation” contains error.
- based on the description on page 10: “The sample with a lower pH value gives a higher surface area (234.273 m²/g). Vice versa, the higher pH give higher surface area (264.276 m²/g)” - it is difficult to tell which pH is responsible for obtaining a higher surface area.
- As for the selection of literature, there are still citing post-conference materials instead indexed and significant scientific in this subject journals
Author Response
The physicochemical properties of organo-silica xerogels derived from organo catalyst were pervasively investigated, including the effect of one-step catalyst (citric acid) and two-step catalyst (acid-base), and also to observed effect of sol pH of organo-silica xerogel toward the structure and deconvolution characteristic. The organo-silica xerogels were characterized by FTIR, TGA and nitrogen sorption to obtain the physicochemical properties. The silica sol-gel method was applied to processed materials by employing TEOS (Tetraethyl orthosilicate) as the main precursor. The final molar ratio of organo-silica was 1:38:x:y:5 (TEOS:ethanol: citric acid: NH3:H2O) where x is citric acid concentration (0.1-10 x 10-2 M) and y is ammonia concentration (0 to 3 x 10-3 M). FTIR spectra shows that the one-step catalyst xerogel using citric acid was handing over the higher Si-O-Si concentration as well as Si-C bonding than the dual catalyst xerogels with the presence of a base catalyst. The results exhibited the highest relative area ratio of silanol/siloxane were 0.2972 and 0.1262 for organo catalyst loading at pH 6 and 6.5 of organo-silica sols, respectively. On the other hand, the organo-silica matrices in this work showed high surface area 546 m2g-1 pH 6.5 (0.07 x 10‑-2 N citric acid) with pore size ~2.9 nm. It is concluded that the xerogels have mesoporous structures which are effective for further application to separate NaCl in water desalination.

Round 3
Reviewer 2 Report
Comment 1:
The introduction is poorly structured. The novelty of the work is not clear. How the article material differs from the publication of the authors [Membrane Technology, Volume 2021, Issue 2, 2021, P. 5-8, https://doi.org/10.1016/S0958-2118(21)00029-X], the reference to which is not given?
Response 1:
To reviewer, thank you very much for the review, advice, correction and suggestion.
We are happy for that and we have d some paragraphs to make it clearer showing the novelty written in the manuscript especially in the introduction.
“In this study, the addition of citric acid catalyst may be a good option. This citric acid (C6H8O7) containing carbon from glucose could be used as an organic catalyst [19]. The advantage of the presence of carbon compound elements in silica matrices is that they able to improve and strengthen the bonds in the matrix [20]. Also, it originally comes from organic acids that are easy to find and cheap. In this study, an inorganic base catalyst, namely ammonia was used. It is to increase the pH of thin films so that pH from thin films is not too acidic. Because of the pH is too acidic it will produce microporous pore size which is not suitable to be applied to water desalination.”
Comment to response 1:
This answer does not provide information about the difference of the material presented in the paper from the material published earlier [Membrane Technology, Volume 2021, Issue 2, 2021, P. 5-8, https://doi.org/10.1016/S0958-2118(21)00029-X], the reference to which is not given. The novelty of the work is still unclear.
Response 2:
To reviewer, thank you very much for the review, correction, and suggestion. The difference between our work and the published article is the utilization of silane precursor. The previous work was employed the dual precursors of TEVS/TEOS, instead in this manuscript is only using single precursor of TEOS. To answer that concern, we already fully revised our introduction section to provide more information about the most recent of our work.
Comment to response 2:
The novelty of the presented work is still unclear. In lines 92-94 authors said “Whereas, the use of single precursor by TEOS and organo catalyst for fabrication organo-silica membrane is not focus investigated”, however in Ref. [Elma, M., Ayu, R., Rampun, E. L. A., Annahdliyah, S., Suparsih, D. R., Sari, N. L., & Pratomo, D. A. (2019). Fabrication of interlayer-free silica-based membranes – effect of low calcination temperature using an organo-catalyst. Membrane Technology, 2019(2), 6–10. doi:10.1016/s0958-2118(19)3003] the system TEOS-citric acid was studied.
Moreover, the title is given in general terms and it is not clear how this article differs from many others on this topic.
Comment 3:
The work contains 35 self-citations among 57 references, which is unacceptable. References are designed carelessly in the text and in the final list.
Comment to response:
The presented version of the article also has problems with self-citation: more than 50% are references to their own works
Response 3:
Dear Reviewer, we are terribly apologised about this issue.
We have removed the several self-citations in the manuscript that is far away related to our work.
Comment to response 3:
Self-citations are beyond common sense in this article, which is unacceptable practice for a journal with a rating like Membranes.
Comment 4:
Extensive English editing is needed because the manuscript contains many typos and grammatical errors. The article is not readable and requires serious revision.
Response 4:
Thank you for the advice.
We have put our best to avoid grammatical mistakes for this manuscript. We regret there were problems with the English. This manuscript has been carefully revised in order to improve the grammatical errors as well as typos. We have also noted the revisions and we have modified the manuscript become much more better.
Comment to response:
Despite the attempts to correct the text, it is far from perfect and requires extensive English editing.
Response 4:
Dear reviewer, we are really sorry about the language issue written in the last manuscript.
We have carefully worked to improve the English in this manuscript, so that this manuscript is totally improved as well as we do avoid the grammatical errors and typos.
Comment to response 4:
Despite the attempts to correct the text, it is far from perfect and requires extensive English editing. There are problems with subjects and predicates, tenses aligning, construction of sentences. This article should be edited by native English speaker.
Other remarks:
Figure 3 does not contain any information.
Line 151: IR spectroscopy cannot be used to study chemical properties.
Conclusions are written in general terms, specific results are not formulated.
Reviewer 3 Report
The authors referred to the indicated comments. The manuscript can be accepted.